# Landslide Susceptibility Mapping Using Machine Learning Algorithm Validated by Persistent Scatterer In-SAR Technique

**DOI:** 10.3390/s22093119

**Published:** 2022-04-19

**Authors:** Muhammad Afaq Hussain, Zhanlong Chen, Ying Zheng, Muhammad Shoaib, Safeer Ullah Shah, Nafees Ali, Zeeshan Afzal

**Affiliations:** 1School of Geography and Information Engineering, China University of Geosciences (Wuhan), Wuhan 430074, China; khanafaq121@cug.edu.cn (M.A.H.); 20161002229@cug.edu.cn (Y.Z.); 2State Key Laboratory of Hydraulic Engineering, Simulation and Safety, School of Civil Engineering, Tianjin University, Tianjin 300072, China; xs4shoaib@tju.edu.cn; 3Ministry of Climate Change, Islamabad 44000, Pakistan; safeershah@uop.edu.pk; 4Chinese Academy of Sciences, Beijing 100045, China; nafeesali@mails.ucas.ac.cn; 5Key State Laboratory of Information Engineering in Surveying, Mapping and Remote Sensing (LIESMARS), Wuhan University, Wuhan 430079, China; zeeshanafzal@whu.edu.cn

**Keywords:** CPEC, random forest, landslides, susceptibility, PS-InSAR, ArcGIS

## Abstract

Landslides are the most catastrophic geological hazard in hilly areas. The present work intends to identify landslide susceptibility along Karakorum Highway (KKH) in Northern Pakistan, using landslide susceptibility mapping (LSM). To compare and predict the connection between causative factors and landslides, the random forest (RF), extreme gradient boosting (XGBoost), k nearest neighbor (KNN) and naive Bayes (NB) models were used in this research. Interferometric synthetic aperture radar persistent scatterer interferometry (PS-InSAR) technology was used to explore the displacement movement of retrieved models. Initially, 332 landslide areas alongside the Karakorum Highway were found to generate the landslide inventory map using various data. The landslides were categorized into two sections for validation and training, of 30% and 70%. For susceptibility mapping, thirteen landslide-condition factors were created. The area under curve (AUC) of the receiver operating characteristic (ROC) curve technique was utilized for accuracy comparison, yielding 83.08, 82.15, 80.31, and 72.92% accuracy for RF, XGBoost, KNN, and NB, respectively. The PS-InSAR technique demonstrated a high deformation velocity along the line of sight (LOS) in model-sensitive areas. The PS-InSAR technique was used to evaluate the slope deformation velocity, which can be used to improve the LSM for the research region. The RF technique yielded superior findings, integrating with the PS-InSAR outcomes to provide the region with a new landslide susceptibility map. The enhanced model will help mitigate landslide catastrophes, and the outcomes may help ensure the roadway’s safe functioning in the study region.

## 1. Introduction

The China–Pakistan Economic Corridor (CPEC) demonstrates the flagship project of the “One Belt, One Road” policy. It is also thought to hold Pakistan’s financial prospects, which are receiving a lot of interest. The Karakoram Highway was built in 1974–1978 and inaugurated in 1979 and runs alongside the CPEC in Northern Pakistan. It is regularly closed for a few months each year because of landslides.

Local topography, tectonic features, geomorphology, landcover, geology and human interference all have an influence on the spatial likelihood of landslides, which is then examined to determine landslide susceptibility (LS) [1]. Landslide vulnerability assessment models frequently assume that historical and current landslide conditions would be constant in the future [2]. LS methodologies can be quantitative or qualitative; quantitative methods evaluate the likelihood of landslide incidence in a susceptible zone, whereas qualitative methods introduce subjectivity into illustrative susceptibility zonation [1,3]. The analytical hierarchy model (AHP) [4,5,6], weight of evidence model [7,8], frequency ratio [4,5,9] and certainty factor [4] are all commonly used landslide susceptibility models. A growing trend is to compare the outcomes of implementing two or more models and the result is a landslide susceptibility model (LSM). However, most studies still use only one model for LSM [1,10]. Reichenbach et al. [1] suggest using numerous models to assess landslides and developing an “optimal” zonation map to reduce risk prediction errors and its integrity to be used for land-use planning. Our literature review of the investigated area demonstrates several statistical approaches for LS such as frequency ratio and weight of evidence [11,12], AHP, and Scoops3D [13], the weighted overlay technique, and the AHP [14] were used in the research region. Several investigations [15,16] provide bivariate analyses that measure the geographical links between particular variables and landslides that influence their occurrence. However, the key disadvantages of these models are that they change the ambiguity of risk processes, are typically static, incorporate geometrical assumptions, and are costly and difficult for the gathering of hydrological and geotechnical data, especially when examining vast and different locations.

In recent years, advances in ML algorithms, computing power, and geospatial innovations have made it easier to create landslide susceptibility (LS) maps [17]. The precision of LS maps can be improved using machine learning algorithms. Knowledge-based methods [18], multivariate logistic regression methods [19,20,21] and multivariate binary logistic regression [22] have all been presented in recent papers. General linear model [23,24], quadratic discriminant analysis [10,24], boosted regression tree [23,25], random forest [26,27,28,29], multivariate adaptive regression splines [30,31], classification and regression tree [23,32], support vector machine [33,34,35], naïve Bayes [36,37], generalized additive model [24,32], neuro-fuzzy and adaptive neuro-fuzzy inference [38,39,40], fuzzy logic [41], artificial neural networks [42,43,44,45,46,47], maximum entropy [48,49] and decision tree [19,50,51] are some of the ML models used in LSM. Qing et al. [52] used various ML techniques for LSM alongside the China–Pakistan Karakoram Highway. In two South Korean catchments, Pradhan and Kim [53] compared the precision consequences of deep neural network (83.71%), and XGBoost (76.73%) approaches for LS mapping. Merghadi et al. [54] assessed the performance and competency of various ML techniques in the literature and discovered that tree-based ensemble optimization algorithms outcompete other ML algorithms. In a comparison analysis, Sahin [55] found that CatBoost had the best precision (85%), followed by XGBoost (83.36%) since the proportion of samples of the model was determined by Catboost was more precisely anticipated than other models. The primary advantages of ML and probabilistic processes are their objective statistical foundation, repeatability, capacity to quantitatively analyze the effect of variables on landslide evolution, and capacity to update them regularly. Machine learning models can be built using a variety of landslide-conditioning factors (slope, aspect, and elevation). Several studies on landslide susceptibility evaluation have been undertaken using remote sensing and GIS techniques [56,57,58].

Furthermore, remote sensing (RS) is an effective method for determining the motion of landslides [59,60,61]. It provides a solution in surveys or enhanced detection in places where catastrophic landslides occur frequently and quickly [62,63,64,65]. Furthermore, interferometric algorithms to radar images effectively map large-scale landslide mapping and detection. It may aid in the development of landslide inventory maps. In particular, decrypt ADInSAR and PS-InSAR [66,67], coherence pixel technique [68], SqeeSAR [69], small baseline subset [70,71], Stanford method for persistent scatterers [72], stable point network [73,74], and interferometric point target analysis [75] have created various useful case research. As noted in past studies [75,76,77], these approaches are involved with mapping and identifying landslide occurrences.

A diversity of researchers in northern Pakistan has analyzed landslides using historical records, field observations, tectonic characteristics, and geological data [78,79,80,81]. Previous studies [14,82,83,84,85] concentrated on probabilistic and statistical relationships and regression interpretation of landslides with parameters. For the first time, the PS-InSAR approach evaluated the surface displacement in the study area using RF, XGBoost KNN, and NB models, making it a distinctive method of identifying landslide movements. Persistent scatterer interferometry (PSI), interferometric synthetic aperture radar (InSAR), and area under curve (AUC) of ROC techniques were used along the KKH to assess displacements and the precision of the models used. Single landslides in hazardous areas can be identified and defined using PS-InSAR. Landslides can also be detected using a spatial statistical method based on a multitemporal assessment of SAR images that calculate slow landslide movements [71].

The current work seeks to develop a susceptibility model and a complete visually interpreted landslide inventory utilizing recently developed ML models, including RF, XGBoost, KNN, and NB. The second goal is to quantify the deformation velocities of slow-moving landslides using PS-InSAR to identify high-susceptibility zones for future landslide disaster management. The third goal is to select the most susceptible model based on accuracy and AUC value and then combine it with PS-InSAR outcomes to produce a new landslide susceptibility map for the research area. These prediction approaches will help lead future development and land management efforts in the area. These susceptibility maps will aid in avoiding and limiting human and economic losses along this critical corridor.

## 2. Methods

### 2.1. Study Area

The research region is 178 km long and has a 5 km radius buffer zone along KKH (Figure 1). The KKH in northern Pakistan is a crucial component of the CPEC; nevertheless, it is frequently disrupted due to several hydro-climatological and geological risks along the route. Landslides are the most common and devastating to highways, human lives, and economic activity.

The research region experiences harsh winters and mild summers. The region’s annual rainfall ranges from 120 mm to 130 mm: the maximum and minimum temperatures vary from 16 °C to −21 °C (Meteorological Department of Pakistan). The lithology of various sources with thicknesses of up to 100 m is irregularly scattered [86,87]. The majority of these deposits are weakly consolidated, making them conducive to landslides in the form of rockfalls and debris flows [80]. The combination of complex topography, high erosion rates, human causes, and active tectonics makes this area one of the most susceptible to landslides.

### 2.2. Geological Setting of the Area

The rocks in the region are mostly Paleozoic, Proterozoic, and Mesozoic in age (Figure 2). According to the geological map prepared by Searle et al. [88], the study area is comprised of the following lithology.

The Chilas complex in the study area comprises mafic and ultramafic plutonic rocks and Kohistan batholiths composed of granodiorite, granite, and diorite. The Gilgit complex metasedimentary rocks are slates, minor phyllite, quartzite, and dolomite limestone. Komila amphibolite comprises of plutonic and meta plutonic rocks with intrusion of diorite granodiorite and granite. In Paleozoic metasedimentary rocks are marble, dolomite, and quartzite.

### 2.3. Landslide Susceptibility Mapping

Geological maps, remote sensing data, and meteorological data were gathered from various sources for the study (Table 1). The Alaska Satellite Facility dataset contained an Advanced Land Observing Satellite, Phased Array type L-band Synthetic Aperture Radar DEM (Digital Elevation Model) with a resolution of 12.5 m (https://search.asf.alaska.edu/ (accessed on 20 January 2022)). Sentinel-2 images with a resolution of 10 m were extracted from the USGS (https://earthexplorer.usgs.gov (accessed on 20 January 2022)) dataset to create a landcover map for the research area. The geological map for the area was digitized in the ArcGIS environment to comprehend surficial geological characteristics. PS-InSAR processing was used to compute the deformation velocity using Sentinel-1 (https://search.asf.alaska.edu/ (accessed on 8 February)) (31 images in descending path and 33 images in ascending path). Figure 3 depicts the approach used in the investigation.

### 2.4. Landslide Inventory

The landslide inventory is the stage in estimating susceptibility since it provides details on all sorts of past landslides in the research area. This is the most important stage since the required precision of the landslide inventory to fine tune the models influences the LSM accuracy [15,89,90]. As a result, the more accurate and high-quality the landslide inventory, the more improved the prediction execution of the SM [89]. The evaluation of landslide hazard begins with creating realistic and detailed landslide inventory maps that show the type of landslide, geographic extension, the date of the event, and location [89,91]. The produced landslide inventory maps are then associated with contributing geo-environmental parameters such as land cover, topography, geology, geomorphology, and other factors to assess the likelihood of terrain causing a landslide allocated to a susceptibility level [1,9,92,93,94].

Inventory maps contain information on all active and historical landslide distributions based on field surveying, aerial image interpretation, and previous report data [80]. In this research, we were using actual data of landslide occurrences obtained from the Geological Survey of Pakistan (GSP) publications [95,96,97], Frontier Works Organization road clearance logs, a research article [98], and Google Earth imagery to produce a multitemporal landslide inventory along the highway. On the other hand, the landslide inventory was created by the visual interpretation of Sentinel-2 photos with 10 m resolution (2020) and from Google Earth and was validated using earlier reports and a field assessment of the research area. Polygon shapes were constructed on satellite images for clearly visible landslides (based on GSP and FWO data). Debris flow (188 locations) and scree slopes (51 locations) are mostly found in the research area as a result of unconsolidated sediments on barren mountains and rainfall, although rock falls (93 locations) are also common as a result of seismic activity and toe cutting of steep slopes by anthropogenic activities for various causes (Figure 4). There were 332 landslides mapped, shown in Figure 5. Of these landslides, 30% (100 landslides) and 70% (232 landslides) were chosen for training for model validation [99].

### 2.5. Landslide Causative Factors

Landslides’ spatial distribution is influenced by triggering, and conditioning variables were chosen based on the region’s morphology, geology, hydrology, and anthropogenic activities. There are no general criteria for choosing independent factors for LSM [71]. The concepts are that the factors must be non-redundant, non-uniform, operational, and measurable [100]. ArcGIS is commonly used to extract important susceptibility conditioning factors from digital elevation models, including elevation, slope, profile curvature, aspect, curvature, and topographic wetness index (TWI) [101]. Land cover, geology, precipitation, roughness, normalized difference vegetation index (NDVI), distance to faults, TWI, slope, plan curvature, curvature, elevation, profile curvature, and aspect were all utilized to estimate the landslides’ disaster susceptibility in the research area (Table 1). All of these maps were converted to a 12.5 × 12.5 m pixel raster format for the models, which was up to digital elevation model resolution. In the resampling method, the cell size of each factor was kept at 12.5 m so that the overlay assessment would obtain the pixels at the same scale, and the output was also the same scale. The maps were digitized at various scales, and the pixel resolution was kept at 12.5 m while converting them to raster format. The DEM with a pixel of 12.5 m was used to extract the majority of the factors, and all other factors were brought to a similar resolution. The thirteen landslide factors are depicted in Figure 6 and Figure 7.

The modeling procedure included machine learning model fitting, identification, and development.

The model unit in this investigation was the grid unit (12.5 m). The spatial resolution of DEM and RS data corresponds to 12.5 m, and all assessment variables have been recalculated at this level.A condition property reached thirteen causative variables and a landslide decision attribute (1 indicates landslides, 0 indicates non-landslides), with each row creating an object.Each column represents an object’s attribute and has been converted into training (70%) and testing the two-dimensional matrix (30%). Training data were used to assemble the models, and test data was used to make forecasts.The landslide susceptibility index maps were created using the forecast values of every model unit per group. The findings of the four algorithms were exported into GIS.The Jenks natural breaks [102] classifications were used to categorize LS: very low, low, moderate, high, and very high. The ROC curve and the area under the ROC curve were used to test the four models.

### 2.6. RF

One of the most widely used methods for regression and classification is the random forest, which was designed by [103]. RF has a lot of important features for classification tasks. Because RF is a non-parametric, non-linear approach, it can handle big datasets with numerical and category data and complicated nonlinearity and interactions between factors. Secondly, it can deal with the situation with more predictors than data and integrate the connection between different predictors. Third, random forest can manage missing values while maintaining precision for missing data.

Furthermore, unlike other ML approaches such as support vector machines and artificial neural networks, RF does not need extensive hyper-parameter tuning. In many instances, utilizing the default parameter values yields good results. When compared to other tree-ensemble approaches (boosting), random forest is relatively fast. Decision trees are built during the training phase, and the output class is based on the classification or regression mode of each individual tree’s decision trees. In order to train random forests, the general method of bootstrap aggregation (also known as “bagging”) is used for tree learners. This bootstrapping approach improves model performance by reducing the model’s variance without raising the model’s bias [104]. Random forest has been extensively utilized for classification applications and large-scale mapping in LSM [26,28] ecology [105], flood mapping [106] and soil science [107].

R statistical software was used to develop the RF model [108]. Because the analysis in the RF model was grid-based, gridded cells (12.5 by 12.5 m) were derived from the randomly shaped sample spatial polygons of landslides and non-landslides, respectively.

The RF model operates by developing numerous decorrelated decision trees as a base learner, with replacement, utilizing a percentage of randomly chosen landslide-predicting variables and landslide observation. Every tree was trained using two-thirds of the randomly chosen training samples, while the remaining one-third of the training samples, called out-of-bag (OOB), was utilized to verify the prediction result. Finally, a pixel was assigned to a class using the majority vote or mode rule [109]. In this research, this model has employed the “randomForest” package in R-studio.

Table 2 lists the three significant parameters: the number of features that are appropriate for dividing (mtry), the minimum number of a sample can also be taken arbitrarily in each bootstrap sample to balance any tree with recursive portioning (ntree) [110].

### 2.7. XGBoost

According to Stanford statistics professor Fridman, the gradient boosting algorithm was developed in 2001 to estimate gradient descent approaches [111]. As the supervised classification model in this work, the XGBoost approach [112] was applied. The approach was invented by the gradient tree boosting algorithm [113,114], which is a powerful machine learning approach. It employs the regularized boosting strategy to prevent overfitting and improve model precision. XGBoost provides scalability for various scenarios, sparse data handling, thorough documentation, minimal computing resource requirements, good performance (i.e., speed), and easy implementation [112]. The approach was chosen since it has won several data science contests [112]. Further adjustments to the approach are needed for extremely unbalanced datasets (e.g., [115]).

Algorithms that boost or lift data are known as “lifting tree models” or “XGBoost”. Their key innovations are summarized below [113].

They optimize their loss function.The candidate split value may be quickly and accurately generated using the parallel approximation histogram method.In addition to a novel sparsity-aware linear tree learning algorithm, they offer an efficient cache-aware block structure for out-of-core tree learning.

In this research, this model has employed the “XGBoost” package in R-studio. Several model preview parameters must be selected for the XGBoost model. User-friendly settings are needed for three of the most important ones: colsample_bytree (column ratio subsamples when each tree is constructed), nrounds (maximum number of iterations boosting), and subsamples (the training instance subsample ratio); (Table 3).

### 2.8. KNN

The KNN algorithm is one of the most fundamental machine learning techniques. It has recently been used in several other disciplines, including LSM [116,117]. KNN uses the k nearest training examples in the components space as input. When it comes to classification difficulties, class membership possibilities describe the degree of uncertainty with which a particular given item may be assigned to any given class [118]. The attributes of the nearby data points are used to classify a data point using a KNN algorithm [119]. It is a more effective version of the ball tree idea [120] that may be used in bigger dimensions. The approach is commonly employed in SM applications [116], and the categorization of a data point’s nearest neighbors determines the chance of it being assigned to any class [118]. The data point chooses the categorization that classifies the greatest number of neighbors. The number of K will be determined through a tuning procedure to obtain better outcomes.

According to Chen et al. [121], they propose that in KNN, objects are evaluated based on the opinions of a majority of their immediate neighbors. The highest consistent closeness of its adjacent neighbors is used to assign the item. If k = 1, the object is solely transferred to the single contiguous neighbor’s class.

### 2.9. NB

NB is a statistical classifier predicated on the Bayesian principle [122]. The Bayes theorem enables this methodology, which is a classification method. The NB maintains that each attribute impacts classification outcomes individually to make estimating the posterior likelihood of observed instances in training data easier [123]. The conditional self-reliance assumption holds that all variables are completely self-sufficient of one another given the output class [124].

The NB technique’s most notable benefits include its robustness to noise and irrelevant variables, ease of use, and lack of reliance on time-consuming iterative procedures [125]. Numerous studies have used the NB approach for LSM [36,37,126]. The following equation can be used to estimate the spatial prediction of landslides using NB:(1)$$y\mathrm{NB}=P\left(yi\right)\overset{n}{\underset{i=1}{\prod }}P\;\left(\frac{xi}{yi}\right)\;\;$$
where *P*(xi/yi) is the conditional probability of each attribute and *P*(yi) is the prior probability of target class yi (landslide).

### 2.10. PS-InSAR

PS-InSAR is an enhanced InSAR technology designed for gradual deformation monitoring or long-term displacement. InSAR is a time series-based method that is broadly classified into two classes: small baseline (SBAS) approaches that focus on spatial correlation and dispersed scattering and PS-InSAR techniques that work on the locations of persistent scatterers (PS) [127]. PSI is a multi-interferometric SAR technique that can estimate ground movement with millimeter precision [128]. The PS-InSAR process uses multitemporal SAR images wrapping the same region to analyze the consistency of the phase and amplitude, which identifies the pixels that are less influenced by spatiotemporal decorrelation and then determines specific deformation details on the constituents of the phase which must be collectively evaluated and modeled to eliminate inconsistencies [129].

We employed Sentinel-1 C-band SAR pictures recorded along both ascending and descending orbit tracks in this investigation. To complete the analysis in C-band data, the PSI [68] requires at least 20 SAR pictures [130]. The PSI monitors surface displacement over months or years, accounting for signal noise, atmospheric, and topographic impacts. This sensor has a ground resolution of around 20 m in the azimuth direction and 5 m in the range direction [131]. This sensor has several acquisition modes, including interferometric wide (IW), wave (Wave), extra-wide swath (EW), and strip map (SM). This study gathered images from the Sentinel-1A IW sensor and analyzed them in SARPROZ software (12 days of temporal resolution). The line of sight (LOS) displacement velocity (V_LOS_) was determined using 0.7 as the coherence threshold in PS-InSAR processing, as shown in Table 4. The InSAR approach computes surface deformation values along the LOS; however, the deformation rate in the LOS direction is inadequate for representing the actual slope displacement [71]. The following equation was used to determine slope velocity (Vslope), which is actually deformation velocity [132]:(2)$$\mathrm{Vslope}=\frac{\mathrm{VLOS}}{\cos\emptyset }\;\;$$
where VLOS is deformation and Ø is the incident angle.

Finally, the calculated result was used for comparative analysis with susceptibility models generated by RF, XGBoost, KNN, and NB methods. The Vslope points were converted into 12.5 × 12.5 grid cells to provide a more precise LSM-like ML model and integrated to enhance the susceptibility degree of those cells defined by ground deformation, minimizing missed alerts, while cells stable and consisting of high susceptibility degrees according to SAR interferometry were not altered [128].

## 3. Results

### 3.1. The Significance of Landslide Variables

To compute the significance of the landslide variables in this study, we utilized R-Studio Software. In comparison, the RF model performed better in estimating the relevance of each element in causing landslides.

Figure 8, using origin software, depicts the significance of the factors using the RF model. The slope and elevation had the greatest impact, according to Figure 8, and profile curvature, roughness, distance to fault, and NDVI were almost equal on landslides in the research region. The slope is critical for landslides in the region (Figure 8); it encourages landslides and makes an area susceptible to landslides. Weathered rocks and medium height frequently define high elevation zones, and slopes are usually overlaid by thin colluvium, making them more prone to landslides [112]. The barren ground is in close contact with climatological factors such as sunlight and precipitation, causing rock deterioration and increasing the likelihood of landslides [133]. Because shear zones and active faults strongly influence landslide activities in the region, the buffer class nearest to the fault line is more susceptible [14]. The bulk of the debris flow, rockfalls, and other slides in the area are caused by monsoon rains [134]. Annual average precipitation data were utilized in this study, which found that while precipitation was not a significant causal factor, there were more landslides in locations with high precipitation (Figure 8). The aspect and plan curvature had a minor impact on the landslide in the studied region. The majority of landslides in the research area are northward facing and south-facing. Arabameri et al. [135] employed RF models for LSM in Iran and found that aspect has a minor impact on LSM. The Komila amphibolite and Gilgit complex metasedimentary rocks are the most vulnerable formations in the study area [11,12,14,101]. The rocks in research area are highly fractured and deformed.

The outcomes of employing the four LSM models obtained using the LPI are depicted in Figure 9. The greater the LPI, the more probable it is that a landslide may happen [136]. The likelihood value of LS was categorized into five classes using the natural break (Jenks) [102] method: very high, high, moderate, low, and very low (Figure 10).

The precision of the maps was evaluated using a confusion matrix, as suggested by [137]. A confusion matrix illustrates the capabilities of the RF, XGBoost, KNN, and NB models during the training stage (Table 5). The RF model shows a high accuracy (0.830) in the research area. Validation was accomplished using the ROC approach [36]. A ROC curve is created in this approach by plotting “sensitivity” versus “specificity” on cut-off values, but it does not fully explain the model’s efficiency; so, the AUC of the ROC curve was utilized to analyze the quantitative functioning of the models [138]. A larger proportion of the area below the curve suggests that the model is more accurate. In contrast, a smaller percentage of the area below the curve shows that the model is less accurate in predicting future occurrences of the phenomena [139]. The AUC of the prediction rate curve was determined to be 88.83, 87.44, 83.38, and 72.80% for the RF, XGBoost, KNN, and NB models, respectively (Figure 11).

### 3.2. PS-InSAR Based Validation

PS-InSAR approaches were utilized to evaluate and verify the models by checking the displacement in the area. The Interferometric Synthetic Aperture Radar (InSAR) approach has been well documented for identifying and tracking mass movements during the previous decade due to its extensive high spatial–temporal resolution, spatial coverage, and operating capacity in all-weather conditions [93]. Many PS-InSAR studies have been conducted to determine the temporal or spatial landslide deformation patterns or the kinematic resolution of slow-moving landslides to quantify the scale of slow-moving landslides [140]. The estimated result was compared to the RF model’s susceptibility model.

The line of sight (LOS) displacement velocity (V_LOS_) was determined using 0.7 as the coherence threshold in PS-InSAR processing (Figure 12). PS-InSAR was also shown to be a useful technique for monitoring slow landslide movement in non-vegetation regions. The InSAR approach computes surface deformation values along the LOS; however, the deformation rate in the LOS direction is inadequate for representing the actual slope displacement [71].

PS-InSAR analysis was performed for both descending and ascending geometries, with VLOS approved deformation in the region. The total number of PS/DS target points acquired with LOS direction deformation results varying from −98 to 73 mm/year was obtained. Using the transformation formula, the VLOS was changed to Vslope. The greatest slope deformation velocity was determined to be −100 mm/year. VLOS indicates just one direction’s deformation based on the satellite’s LOS, which is determined to evaluate slope orientation velocity (Vslope). Because most landslides or ground surface displacements occur along the direction of steep terrain in the event of landslide assertion, Vslope is the main ingredient employed to define landslide advancement. The calculated Vslope for ascending and descending pathways was added together (Figure 13). The only displacements in RF’s highly sensitive zone-produced susceptible model were depicted in an ultimate deformation map (Figure 14). The PSI findings revealed that most of the mapped landslides were in deforming zones, although slow-moving landslides were forecasted more precisely because of the Sentinel-1A sensor’s extended revisiting period.

Finally, the RF-based LSM was combined with Vslope to improve the region’s precise susceptibility map. The Vslope points were converted into 12.5 × 12.5 grid cells to provide a more precise LSM-like RF model and integrated to enhance the susceptibility degree of those cells defined by ground deformation, minimizing missed alerts, while cells stable and consisting of high susceptibility degrees according to SAR interferometry were not altered [128]. The contingency matrix was used to improve an LSM for the region to a Vslope and RF-based susceptibility model (Figure 15). In other words, the degree of difference for each cell was evaluated using the newly created LSM, which was generated using the RF model.

## 4. Discussion

The findings demonstrate that the RF, XGBoost, KNN, and NB data-mining techniques have comparable precision for LSM along the KKH, with the RF outperforming the others in terms of AUC value and accuracy. Our results conform to the consequences outlined in other work [11,12,14,52,101].

The overall precision values found in this research (RF, 83.0%) were compared to Youssef et al. [23]; it was discovered that the precision values found in this study were higher than the RF (81.2%) in other ML models of the revealed research. Sevgen et al. [29] compared ANN, logistic regression, and RF for LSM and found that the RF model shows the best classification precision with respect to ANN and LR. Taalab et al. [26] evaluated the RF algorithm for landslide in northwest Italy and found that the RF model performed well compared to other tree-based models. Chen et al. [141] reassembled the random forest (RF), logistic model tree (LMT), and classification and regression tree (CART) models to map LS. The LMT (74%) and CART (73%) models showed slightly lower precision values than the RF model (77%); RF performed better in LSM. Zhang et al. [142] demonstrated that the random RF model outperformed the C5.0 decision tree model by comparing it to the C5.0 decision tree model. The RF technique has an advantage over other ML models. It can use multiple input parameters without removing them and provide a limited number of classes with good forecast precision [143]. This model’s categorization precision is determined by the training dataset’s type, scale, number, and accuracy. The combination of all appropriate parameters boosts the precision of this model. Furthermore, compared to other models, RF has a greater capacity to implement a large number of data [144]. Arabameri et al. [135] employed RF models for spatial modeling of gully erosion in Iran and found that RF performed best. Zhang et al. [145], when compared to neural networks, decision trees, and the RF, obtained the best results for debris flow susceptibility with the RF method in Shigatse Area, China. In areas with debris flow and rockfalls, we discovered that the RF model is the best predictive technique for LSM.

Recently, InSAR approaches for producing and updating landslide inventories have been created [146]. The findings of the InSAR techniques are thought to be more precise [147], yielding susceptibility maps with high production precision. The LOS velocity statistics only reveal the velocities along the slopes in the highly sensitive zones of both models. PS-InSAR was also shown to be a useful technique for monitoring slow landslide movement in non-vegetation regions. The ROC curve AUC was used to evaluate prediction capabilities, and it predicts 88.83, 87.44, 83.38, and 72.80% for RF, XGBoost, KNN, and NB, respectively, confirming the model’s accuracy. The collected susceptibility map was categorized into five groups using the Jenks natural breaks [102]: very low, low, moderate, high, and very high. In comparison, the RF model performed better in estimating the relevance of each element in causing landslides.

LSM was performed using the RF, XGBoost, KNN, and NB in this work. Nonetheless, several limitations caused misclassification in the results, such as (1) the accuracy of the landslide inventory and (2) the accuracy of data connected to each landslide variable. Because of the severe environment along KKH, only 332 landslides have been mapped for this research region. It resulted in considerable misclassification inaccuracies for the LSM, emphasizing the significance of upgrading the LSM utilizing PS-InSAR results. Surprisingly, when paired with the PS-InSAR data, the novel LSM reduced misclassification in which landscape altered by slope deformity was categorized as extremely low and very low.

Another problem is that the landslide susceptibility mapping merely represents anticipated landslide dispersion in regions rather than interactive displacement processes through time. Variations in landslide behavior over time, on the other hand, are a serious challenge for decision-makers [148]. In conjunction with the PS-InSAR outputs, a new landslide susceptibility map can depict the real conditions of landslides. It can be designed for quantitative hazard assessment and preliminary landslide mapping at the province level [149].

The LSM creates a susceptibility map for landslides, identifies the important variables that cause landslides, and evaluates the effect and their contribution [27,150]. Land cover, geology, slope, precipitation, NDVI, distance to faults, elevation, curvature, plan curvature, TWI, profile curvature, roughness, and aspect were all utilized to estimate the probability of landslides disaster in the research area. The main contributors of landslides in the area are slope, elevation. The slope is critical for landslides in the region (Figure 8); it encourages landslides and makes an area susceptible to landslides. Weathered rocks and medium height frequently define high elevation zones, and slopes are usually overlaid by thin colluvium, making them more prone to landslides [112]. Because shear zones and active faults strongly influence landslide activities in the region, the buffer class nearest to the fault line is more susceptible [14].

Previous studies such as [13] in this area relied mainly on statistical models, and a considerable number of landslides were missing in the inventory. As a result of the inadequate landslide inventory, the LSM is ineffective. This work focused on complete mapping of landslides to identify primary landslide triggers and define high susceptible zones using the PS-InSAR approach, which will be used in the future to mitigate landslide risks in the region.

## 5. Conclusions

Landslides are one of Pakistan’s most devastating natural disasters, generating major risks to lives and socioeconomic damage each year. So far, the process of landslide mapping has been highly difficult and volatile to perform correct and quick estimation of landslides in most places. Multiple attempts have been made to improve reliability based on many forecast models for mapping the landslide susceptibility, targeting different locations for this goal. Decision-makers must construct more relevant landslide susceptibility maps to improve the prediction model’s performance. This study concentrated on complete landslide mapping to determine the fundamental causes of landslides and designate high-risk zones, which will be useful in the future to mitigate landslide threats in the region. The study’s distinctive feature is that it provides more accurate LSM by employing ML models verified by PS-InSAR processing.

This study used RF, XGBoost, KNN, and NB ML algorithms to enhance the LSM of the Karakoram Highway using the PS-InSAR approach. This study assessed the vulnerability using elevation, precipitation, slope, land cover, roughness, NDVI, curvature, distance to faults, plan curvature, aspect, profile curvature, geology, and TWI. Slope, elevation, and profile curvature are the primary causes of landslides in the region. The susceptibility model created will be used to identify zones for construction growth and improved management planning along the KKH. The LSM illustrates the just forecast landslide distribution in regions, not the dynamic displacement process over time. Variations in landslide activity eventually, on the other hand, are a main consideration for decision-makers. The newly developed LSM, when merged with the PS-InSAR results, may show the true situation of landslides and should be utilized for quantitative hazard analysis and preparatory landslide mapping at the regional level. Geotechnical and other slope stabilization procedures are necessary to minimize future landslide catastrophes in an environment. We conclude that our approach can give valuable insights into highway safety measures.

## Figures and Tables

**Figure 1 sensors-22-03119-f001:**
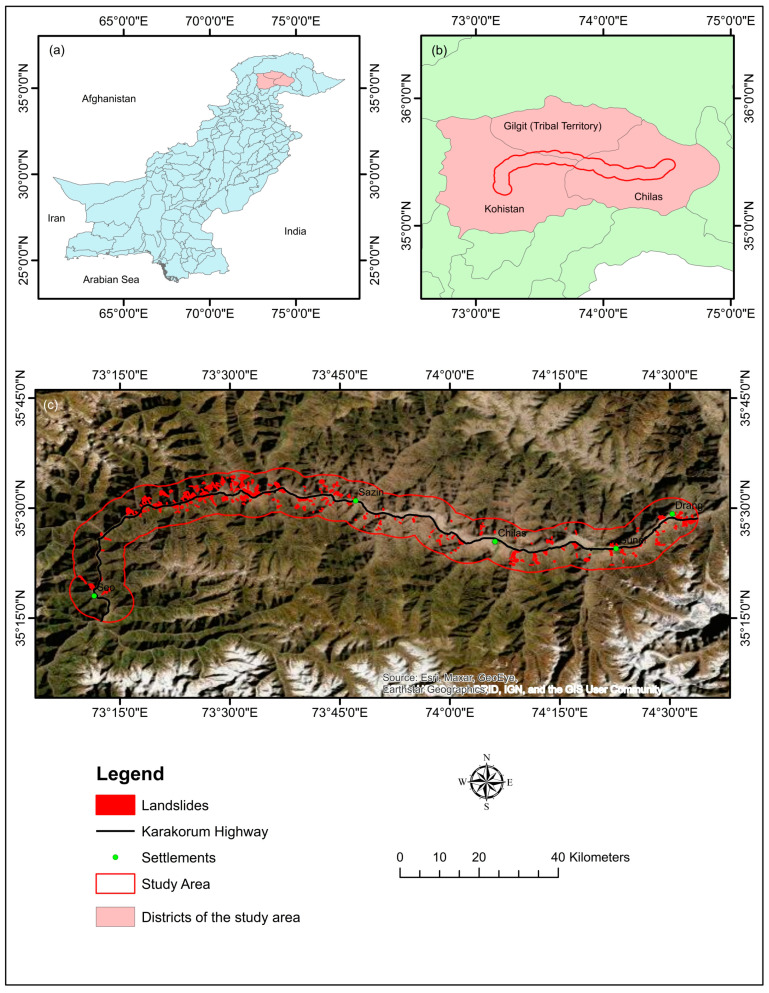
The 541 band combination Landsat image showing the area under investigation. (**a**) Pakistan, (**b**) District boundaries, (**c**) Study area in red outline.

**Figure 2 sensors-22-03119-f002:**
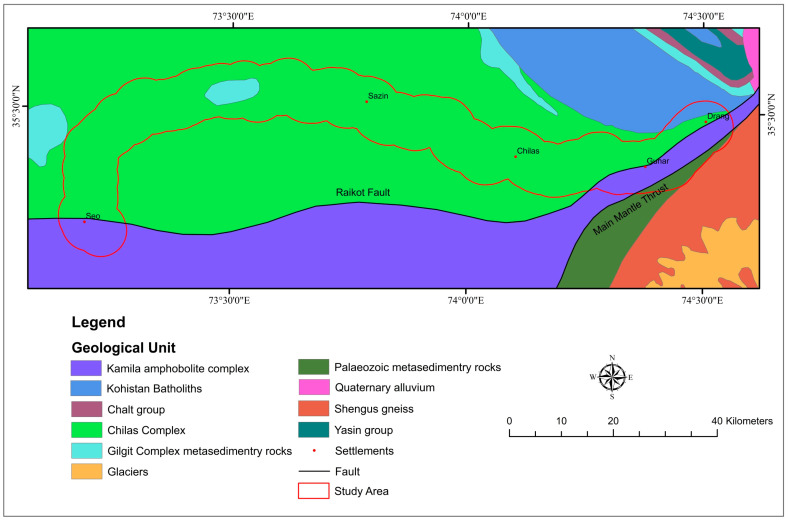
Regional geological map of the study area.

**Figure 3 sensors-22-03119-f003:**
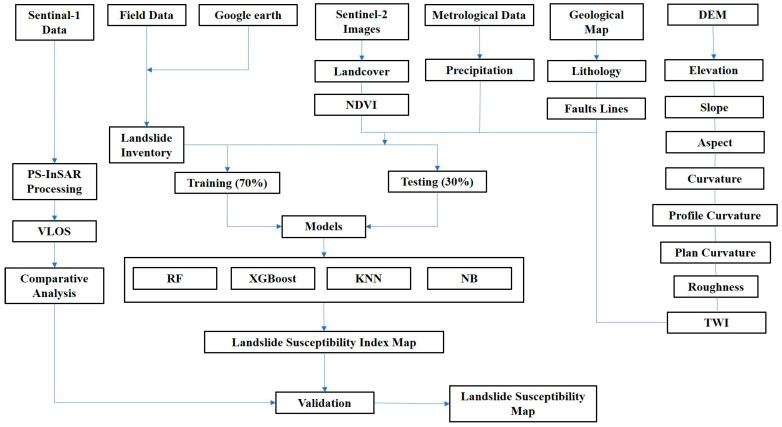
Flow chart of research.

**Figure 4 sensors-22-03119-f004:**
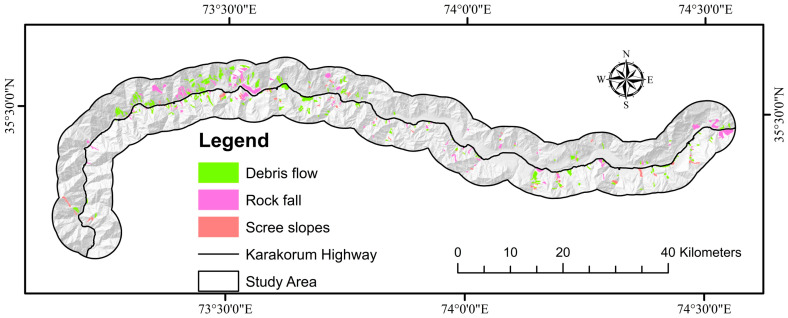
Showing the debris flow, rockfall, and scree slopes in the research area.

**Figure 5 sensors-22-03119-f005:**
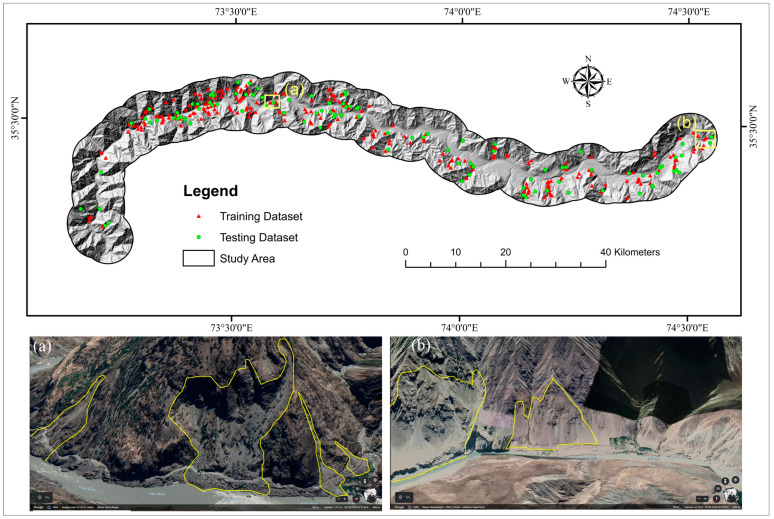
Landslide inventory of study area. (**a**) Shows the rockfall and scree slopes near the Sazin area, (**b**) rockfall and debris flow near the Drang area.

**Figure 6 sensors-22-03119-f006:**
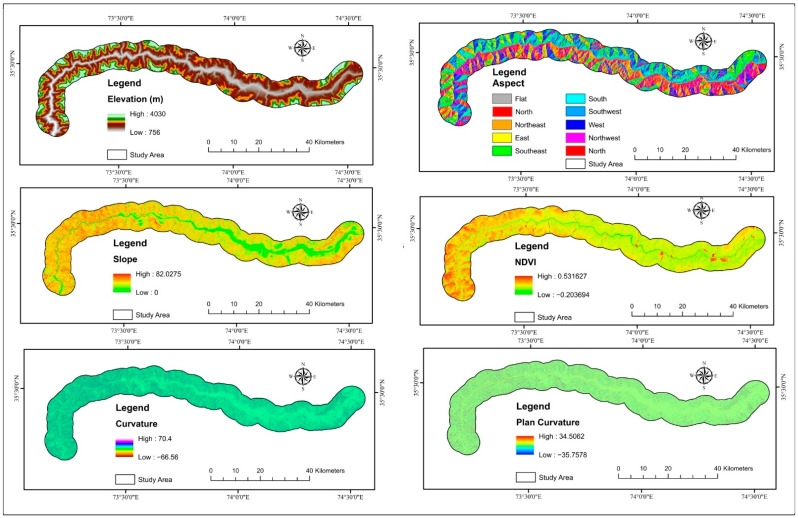
Landslides factors used in the research study.

**Figure 7 sensors-22-03119-f007:**
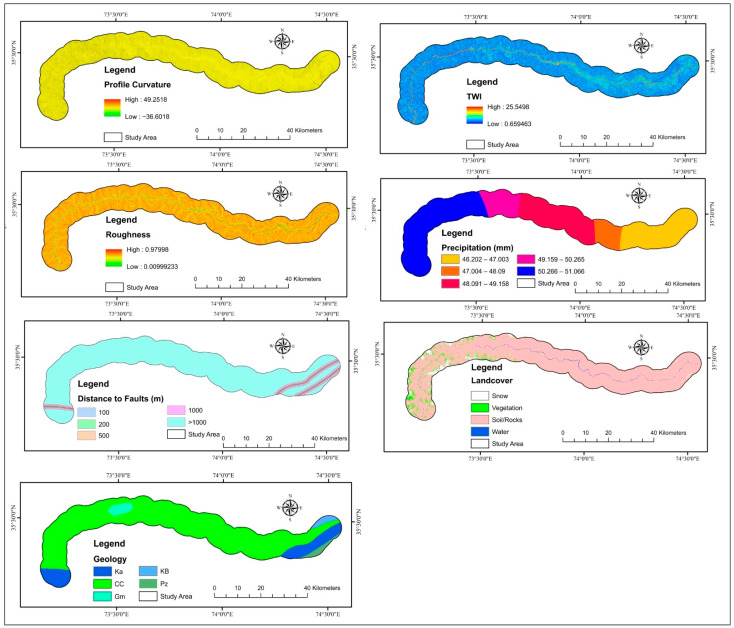
Landslides factors used in the research area.

**Figure 8 sensors-22-03119-f008:**
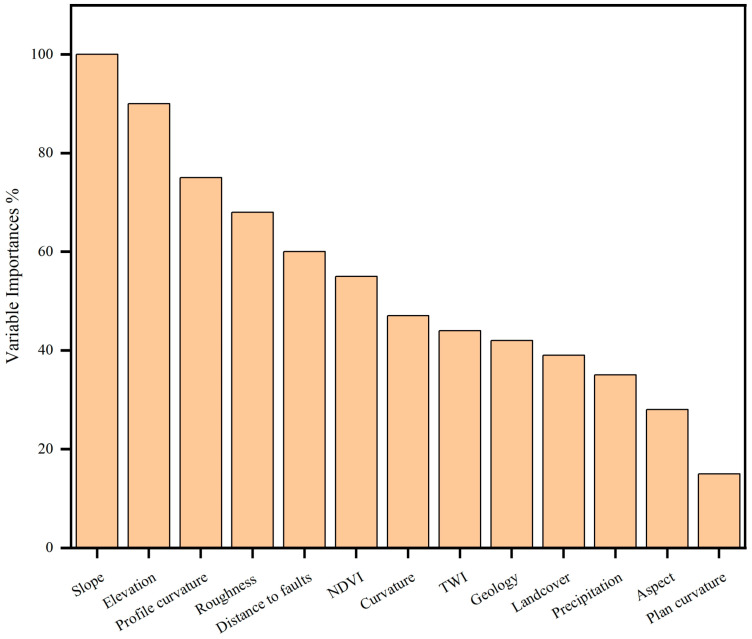
Factors important in the research region using the RF model.

**Figure 9 sensors-22-03119-f009:**
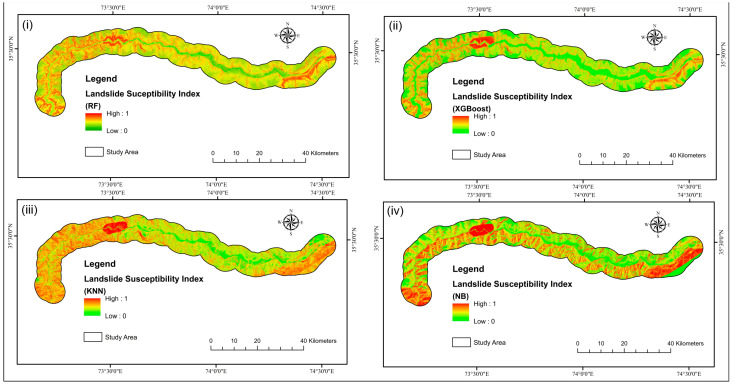
Landslide susceptibility index maps (**i**) RF, (**ii**) XGBoost, (**iii**) KNN, (**iv**) NB.

**Figure 10 sensors-22-03119-f010:**
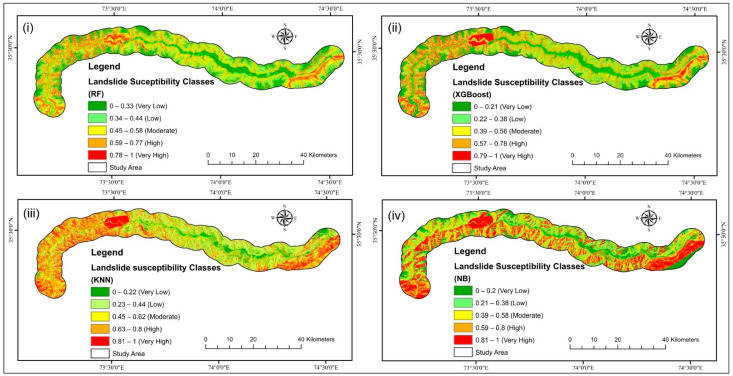
LSM using (**i**) RF, (**ii**) XGBoost, (**iii**) KNN, and (**iv**) NB models.

**Figure 11 sensors-22-03119-f011:**
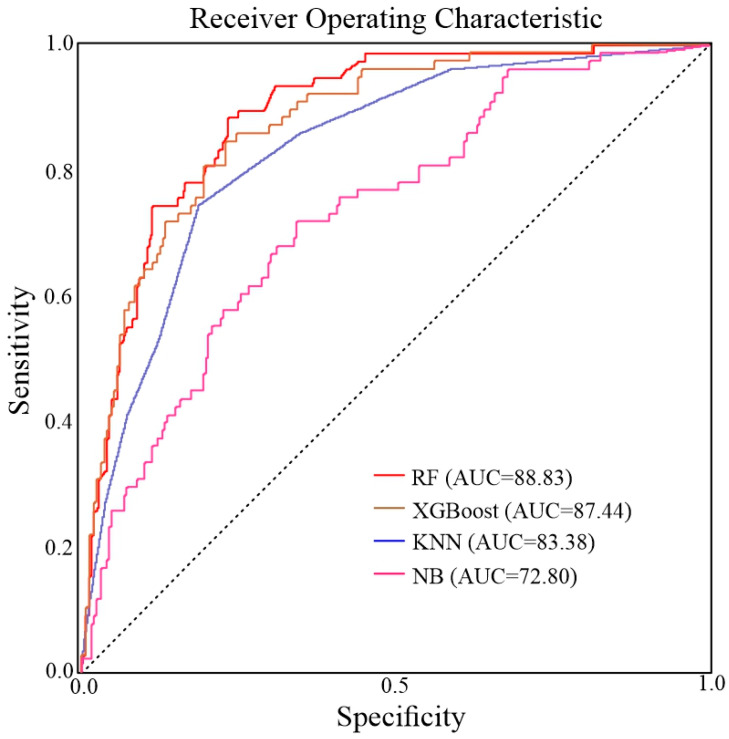
Receiver operating characteristic plots of models.

**Figure 12 sensors-22-03119-f012:**
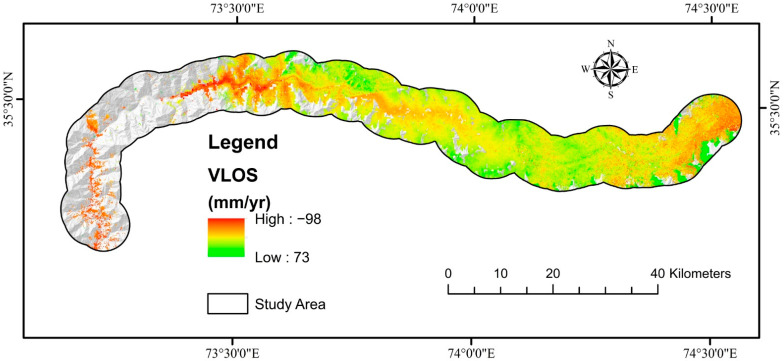
Landslide deformation velocity map along LOS direction for ascending and descending paths using PS-InSAR.

**Figure 13 sensors-22-03119-f013:**
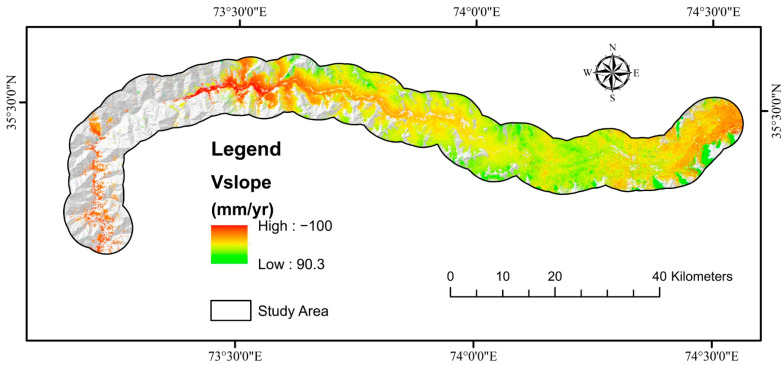
Showing the deformation velocity along slope direction for both ascending and descending paths using PS-InSAR.

**Figure 14 sensors-22-03119-f014:**
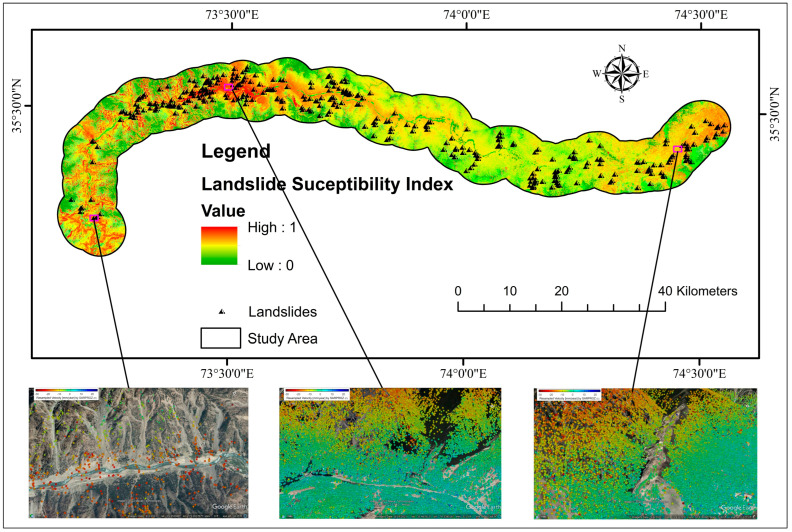
PSI distribution of LOS deformation velocity on Google Earth using RF landslide susceptibility index.

**Figure 15 sensors-22-03119-f015:**
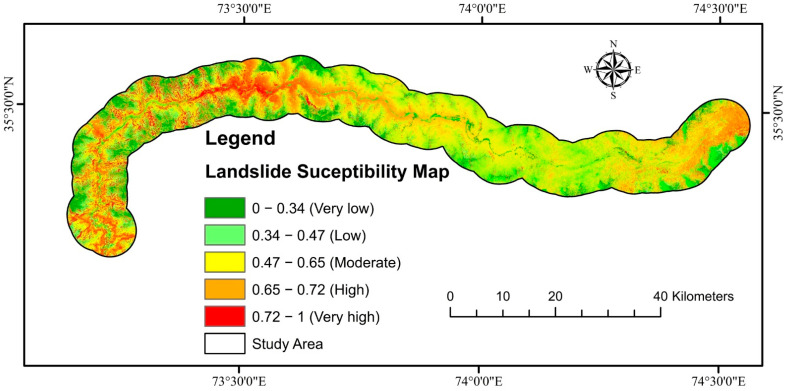
Final LSM via Vslope.

**Table 1 sensors-22-03119-t001:** Information on landslide conditioning factors.

S.NO	Factors	Description/Extraction	Category
1	Elevation, aspect, curvature, slope, profile curvature, TWI, plan curvature, roughness	ALOS-PALSAR DEM (https://search.asf.alaska.edu/ (accessed on 20 January 2022))	Topography
2	Geology, distance to fault	Geological Survey of Pakistan	Geology
3	Landcover	Land cover classes (https://earthexplorer.usgs.gov) (accessed on 20 January 2022) (Sentinel-2 images)	Conditioning factor
4	NDVI	Normalized Different Vegetation Index (Landsat-8, 2021)	Landcover
5	Precipitation	Annual rainfall (Pakistan Metrological Department)	Triggered factor

**Table 2 sensors-22-03119-t002:** Parameters used in RF model.

Parameters	Values
Node size	14
mtry	10
ntree	500

**Table 3 sensors-22-03119-t003:** Parameters used in XGBoost model.

Parameters	Values
nround	210
colsample_bytree	1
subsample	1
max_depth	6
eta	0.05
gamma	0

**Table 4 sensors-22-03119-t004:** Details of PS-InSAR processing.

Specification	Ascending	Descending
Temporal range	1 May 2020–20 May 2021	14 May 2020–9 May 2021
No. of images	33	31
No. of PS/DS	526,815	450,990
Minimum VLOS (mm/year)	−98	−34
Maximum VLOS (mm/year)	31	73

**Table 5 sensors-22-03119-t005:** Confusion matrix of models.

Models	Observation	Predicted		Accuracy
		No	Yes	
**RF**	No	35	12	0.830
	Yes	43	235	
**XGBoost**	No	33	13	0.821
	Yes	45	234	
**KNN**	No	32	18	0.803
	Yes	46	229	
**NB**	No	39	49	0.729
	Yes	39	198	

## Data Availability

The data presented in the study are available on request from the first and corresponding author. The data are not publicly available due to the thesis that is being prepared from these data.

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
