# Peer review of "Landslide Susceptibility Mapping Using Machine Learning Algorithm Validated by Persistent Scatterer In-SAR Technique"

_sensors, 2022, doi:10.3390/s22093119_

Round 1

Reviewer 1 Report

Dear Authors,

I just finished reviewing the manuscript entitled “Landslide Susceptibility Mapping Using Machine Learning Algorithm Validated by Persistent Scatterer In-SAR technique”

The authors attempted to predict landslides occurrence using different machine learning algorithms and validated by Persistent In-SAR technique. The work has been achieved with good efforts. However, the manuscript suffers several problems in language and construction

General comments

There are several typos and grammar mistakes. So, the manuscript should be passed to a native speakers and proofread

The abstract

The abstract needs to be reworded by focusing on the method and main findings

Methods

The section is not clear and written in a bad manner. More details about the data source and methods are needed. How does the author constructs the LS conditioning factors?

The method of validation by Persistent Scatterer In-SAR technique is missing in the manuscript

Result

The result is mixed with the method and not clear

I highlighted some comments on pdf versions

Based on these comments, the manuscript should be rejected and resubmitted

Reviewer 2 Report

Authors conducted this research in the title of “Landslide Susceptibility Mapping Using Machine Learning Algorithm Validated by Persistent Scatterer In-SAR technique".

The paper’s subject could be interesting for readers of journal. Therefore, I recommend this paper for publication in this journal but before that, I have a few comments on the text that should be addressed before publication:

Comments:

  • Line 323: In line 323 authors used this word “We” at the beginning of the sentence. The beginning of a sentence is a noticeable position that draws readers’ attention. Thus, using personal pronouns as the first one or two words of a sentence will draw unnecessary attention to them, so it is better to avoid starting a sentence with personal pronouns.
  • Abstract: In the Abstract section, authors should pay more attention to the main goals and questions that are supposed to be addressed in this article. It would be really helpful for readers of this paper because they first read abstracts to know if an article interests them or is related to a subject important to them. Instead of checking numerous written materials, readers depend on abstracts to quickly determine if an article is relevant to them or not.
  • Conclusion: All research papers should contain a funding acknowledgement statement included in the manuscript in the form of a sentence under a separate heading entitled “Research funding” (or similar phrases) directly after the “Acknowledgements” and “Declaration of Conflicting Interests”, if applicable, and prior to References.
  • Which software has been used in this work to export the charts and diagrams in this work? For instance, software like SigmaPlot or SmartDraw are used to export and depict charts. Mentioning used software would be helpful to future researches and studies in the field of this article.
  • Line 401: The equation has been used in this line without any number related to it. Every equation should have its own number. In addition, there could be more space between equations and lines.
  • Conclusion: In the conclusion section authors should mention more words about their suggestions to future works. It really can be helpful for future studies and works related with title of this article. For example, addressing limitations of your research, your research will not be free from limitations and these may relate to formulation of research aim and objectives, application of data collection method, sample size, scope of discussions and analysis etc. You can propose future research suggestions that address the limitations of your study.
  • Table 3: The title of this table could be moved to the right direction and aligned in the middle. It looks better and seems more integrated with other title of tables in this article.
  • Keywords: “ArcGIS“ could be mentioned in this section because it has been used repeatedly in different parts of this paper and it seems a keyword.
  • Figure 6: Some utilized numbers and letters in this figure look blurry and it is really hard to read them. Every number and letter used in this research should be clear and easy to read.
  • Since recently it has been proved that artificial intelligence (AI) and machine learning has a numerous applications in all of engineering fields, I highly recommend the authors to add some references in this manuscript in this regard. It would be useful for the readers of journal to get familiar with the application of AI in other engineering fields. I recommend the others to add all the following references, which are the newest references in this field

[1] Lotfi, F., & Semiari, O. (2021, April). Performance Analysis and Optimization of Uplink Cellular Networks with Flexible Frame Structure. In 2021 IEEE 93rd Vehicular Technology Conference (VTC2021-Spring) (pp. 1-5). IEEE..

[2] Roshani, M., et al. 2020. Application of GMDH neural network technique to improve measuring precision of a simplified photon attenuation based two-phase flowmeter. Flow Measurement and Instrumentation, 75, p.101804.

[3] Lotfi, F., Semiari, O., & Saad, W. (2021). Semantic-Aware Collaborative Deep Reinforcement Learning Over Wireless Cellular Networks. arXiv preprint arXiv:2111.12064.

[4] Charandabi, S. E., & Kamyar, K. (2021). Prediction of Cryptocurrency Price Index Using Artificial Neural Networks: A Survey of the Literature. European Journal of Business and Management Research, 6(6), 17-20.

Round 2

Author Response

The manuscript was proofread by a native speaker. And the point to point changes have been made according to the suggestions given in the pdf file.

Reviewer 2 Report

All comments have been addressed correctly

Author Response

All comments have been addressed correctly.

Round 3

Reviewer 1 Report

The new version looks very good and I recommend it to be published in its current form